# Sediment Resuspension Distribution Modelling Using a Ship Handling Simulation along with the MIKE 3 Application

**Jure Srše ***, **Marko Perkovič** 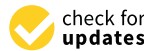 **and Aleksander Grm** 

Faculty of Maritime Studies and Transport, University of Ljubljana, Pot Pomorščakov 4, 6320 Portorož, Slovenia; marko.perkovic@fpp.uni-lj.si (M.P.); aleksander.grm@fpp.uni-lj.si (A.G.)
* Correspondence: jure.srse@fpp.uni-lj.si

**Abstract:** The environmental effects of ship propellers were not even close to fully examined before the current massive ships were introduced to sea trade. Larger ships, result in greater length, beam, draft and propulsion power. Of concern here is the under-keel clearance (UKC) and applied power, the most important parameters causing sea bottom sediment resuspension and, consequently, the transport and deposition of washed sediments. The problems are multifarious: shorelines could be contaminated with heavy metals, petroleum hydrocarbons and other organic chemicals, which are sometimes buried deep in the sediment bed. The effects of resuspension on marine life have been well documented by marine biologists. Further, a ship passing through a flow field may have a significant hydrodynamic effect on the shipping channel: waves generated by moving vessels can accelerate shoreline erosion; erosion around quay piles have a negative impact on sea flora. Waves can also affect other manoeuvring vessels or ships at berth. Available empirical models are applicable for a steady state condition, addressing velocity and, consequently, shears at the sea bottom for defined UKC and states of applied power. The idea here is to calculate material resuspension dynamically in the water column based on realistic manoeuvring conditions, which can be a matter of some complexity. During a manoeuvre, the pilot must bring the ship into or out of the harbour in the safest possible way, operating the telegraph, rudder, thrusters and possibly tugs, and also co-ordinating the work of the linesmen. The jet speed powering the vessel is not only a function of the speed of the propeller, but also of the present speed of the ship, which has an effect on the propeller's constantly changing torque. Additionally, the bathymetry is constantly changing, and the streamlines hit not only the seabed, but also the bank and other structures of the harbour basin. The resuspended material remains in the column long after the ship has finished manoeuvring, moving slowly through the entire water column and being transported not only by the remaining streamlines of the ship but also by general currents. Realistic manoeuvring parameters can be obtained from real-time simulations with a real crew using state-of-the-art Full Mission Bridge Simulators (FMBS); eddies and the like contribute to the distribution and material resuspension and can be calculated by applying numerical modelling. In this paper, a container ship departure manoeuvre is simulated dynamically using Wartsila FMBS obtained data, which are postprocessed and coupled with the MIKE 3 FM hydrodynamic modelling application to which we add the precise port of Koper bathymetry to gain ship propeller spatial jet velocity distribution in specific time domains. Obtained jet velocity distribution is further coupled with the MIKE 3 MT particle tracking application to visualize total resuspended sediment transport patterns, etc. Container ships were selected to amplify the urgency of this phenomenon; they are the most intrusive in terms of resuspending and scouring the seabed given their powerful engines and larger propellers. Passenger ships could have been used, car carriers, or even tankers; but the fear among scientists is that the issue will not be taken seriously enough by certain stakeholders.

**Keywords:** sediment resuspension; propeller jet velocity field distribution; full mission bridge simulator; MIKE 3 FM



## 1. Introduction

The profitability of larger ships is blinding the community of maritime commerce to the numerous problems that form a chain trailing from the inexorable drive to unquestionably float ever larger ships. Ports around the world are struggling to accommodate deep draft vessels, and though this paper studies the northern Adriatic near the Port of Koper, which sees more than its share of pollution, we are also keenly aware of the safety issues caused by the premature predominance of overly large vessels (collisions, groundings, near misses, delays, increased effects of weather). The trend of the future is toward ships that call at ports with low under-keel clearance (UKC), with less and less distance between the seabed and the propeller tip and greater main engine and thrusters' power. Propeller thrust, propeller diameter and the other factors mentioned above create a propeller jet velocity field that results in sediment resuspension (SR), which has known negative effects on sea flora and fauna, berth walls, quay piles (erosion) and the deposits of resuspended sediment [1]. This paper deals with sediment resuspension in the Bay of Trieste. This location is specific because of the Idrijca River, which deposits mercury in this region [2]. The process of sediment resuspension by ship propellers raises mercury particles that enter the marine food chain and, interestingly, deliver some mercury to the very mouth of the river that flows into the gulf.

The study area is shown in Figure 1, where three basins of the port area are visible; the container terminal is located in the first basin, close to the city centre.

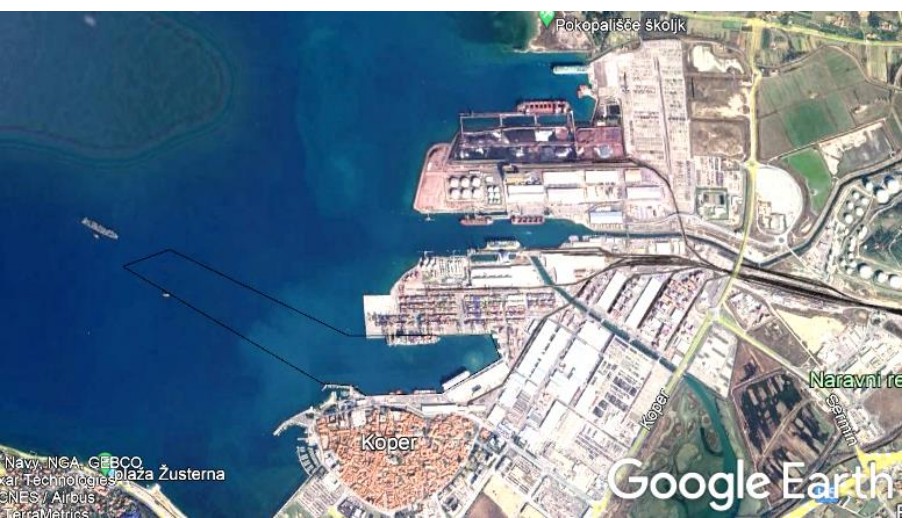

**Figure 1.** Black line presents studied area; Basin I of the harbour of Koper, Slovenia [Google Earth image].

The velocity field of the propeller jet reaches the seafloor at some distance from (behind) the vessel, depending on the sea depth and the topography of the seabed. Critical jet velocity is the speed at which the particles overcome their own shear stress and resedimentation occurs; obviously, this varies according to the particle characteristics. Additionally, metocean conditions such as currents, density gradients and prevailing winds affect sediment transport and deposition. After the resuspension process, sediment much of which is too dispersed to be tracked and microparticles may travel some miles before settling is deposited in visible masses at predictable locations within the port basins. The bathymetry of the port is constantly changing (this is darkly ironic, as it may require dredging where at much expense dredging was required that the port might accept these very ships, as will be mentioned) and may prevent vessels from docking and manoeuvring at a particular point due to insufficient UKC. Port authorities must perform dredging to allow deep draft vessels to safely enter and berth in the port. The work is time-consuming, causes congestion in the port and creates additional costs due to demurrage costs. The first step in finding a solution to minimize SR is to understand the velocity distribution of the propeller jet. Tools

to evaluate this thrust include the Wartsila NTPro 5000 v 5.35 full mission bridge simulator, and now the introduction of hydrodynamic and mud transport models MIKE 3 FM and MT [3].

Recent empirical modelling studies have included ship propeller jet distribution in water columns at specific distance from propeller orifices, but not effects on sea bottom sediment resuspension [4]. Some existing approaches have been limited to coupling resuspended material transport in specific time domains, sea bed roughness, bathymetry, port basin shape and metocean impacts on sediment transport [5].

Several authors performed laboratory experiments to validate propeller rotation impact on the sea bottom. The research aim of the author [6] was to determine scouring patterns based on twin propeller configuration. The measurements were done with 2D Laser Doppler Velocimetry to determine scouring patterns in x and y directions. The limitation of the research is the two-propeller set-up, both left turning; vessels in reality have opposite turning propellers due to elimination of propeller side walk. The authors [7] investigated maximum scour depth, width, length and deposit height, with experimental (Dp = 8.2 cm) ship four blade propeller. Their experiment included three different propeller axial distances from the bed and three propeller rotational speeds. They scanned the bed surface with the photogrammetry technique combined with a Terrestrial Laser Scanner (TLS). The main contributions of this paper were that propellers generate two holes; one smaller directly under the propeller and one bigger downstream. The propeller scouring is not symmetrical in a lateral direction but is deeper on one side with higher resuspended particles deposition on the other side. These authors [8] presented experimental results of the propeller jet flow field around an open quay using the Particle Image Velocimetry (PIV) technique. The experiment included four longitudinal distances from the propeller face to quay toe slope. The results showed that maximum scour depth appears at the toe of the slope and decreases with longitudinal distance from the propeller face. This study was useful for understanding and predicting erosion processes in narrow channel approaches and along port quay infrastructure. The authors presented ship propeller scour patterns using an experimental set-up and stationary position of the propeller. An upgrade could be repeating the work with a propeller moving through the ambient water. Results in this research could be used only in the first step of a ship departure manoeuvre, when ship velocity is zero or close to it.

## 2. Tools and Methods for the Determination of Ship Propeller Jet Velocity Field Distribution

There are many ways to obtain the kinematic parameters of a ship's motion. The best would be from the vantage of the ship's team onboard taking VDR data; but of course this is logistically complex. Another option is to reproduce the manoeuvre using the state-of-the-art ship handling simulators operated by actual marine pilots, which is what we did. The simplest option which is insufficiently accurate is to use Automatic Identification System data transmitted by every vessel in the area.

### 2.1. Automatic Identification System and Full Mission Bridge Simulator

The automatic identification system (AIS) is a standard tool in the maritime industry, primarily intended for easy identification of ship traffic. Other applications of AIS include collision prevention, emission awareness, tracking oil spills, monitoring fishing activities, spatial planning and much more [9]. Data transmitted by AIS is used in scientific research because the system provides a large amount of statical and dynamic data: vessel name, MMSI number and call sign, type and dimension of vessels, vessel position, course over ground, speed over ground, heading; from this data distance and time of closest point of approach can be determined with any particular object in the vicinity. Extensive data from AIS were analysed for the port of Koper, Basin I, where container ships berth. The decision for selecting this particular location was taken regarding the high container ship traffic density (which is expected to grow in the near future) and low UKC, as little as 0.5 m. Based on acquired AIS data, a complete sailing departure route leg can be reproduced in

the simulator. According to the preliminary analysed data, the outbound manoeuvre of a container ship was selected and simulated; in order to simplify the first modelling, only the propulsion system of the main engine was analysed, without tug assistance and bow thruster. The following table (Table 1) contains the relevant data from simulator container ship 1.

**Table 1.** Main particulars of subject vessel.

| Ship Particulars | Value/Type |
| --- | --- |
| Length Overall (*LOA*) | 203.6 m |
| Moulded Breadth (*B*) | 25.4 m |
| Summer Draft ($T_s$) | 9.8 m |
| Displacement (*D*) | 32,025 t |
| Max engine power (*P*) | 15,890 kW |
| Ship propeller type | Fixed Pitch Propeller (FPP) |
| Propeller diameter ($D_p$) | 6.0 |
| Propeller immersion | 6.67 m |
| Bow thruster capacity | 763 kW |

The figure below (Figure 2) presents the procedure for obtaining dynamical data of a container ship departure manoeuvre needed for further numerical modelling with MIKE 3. The FMBS simulation program was used to reproduce the departure manoeuvre in real time and place.

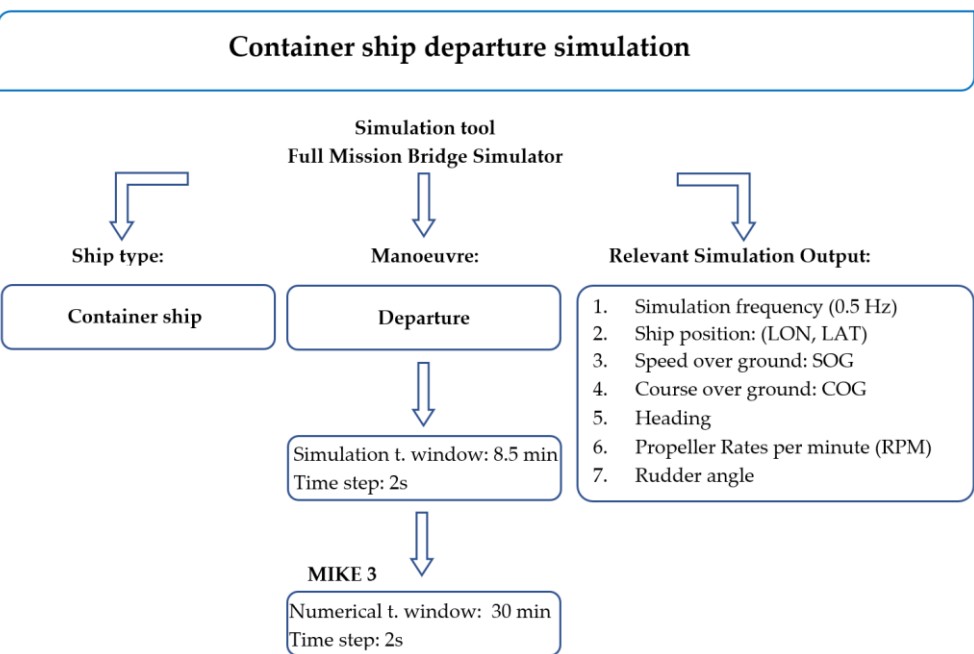

**Figure 2.** Study flow chart for the acquisition of dynamic data of a container ship departure manoeuvre and numerical modelling.

Figure 3 shows a screenshot of the simulator instructor station, showing the harbour basin 1 with high resolution bathymetry and the position of the ship at berth. During the manoeuvre, relevant ship dynamic data were recorded at a frequency of 0.5 Hz: vessel position (lon., lat.), speed over ground (SOG), course over ground (COG), heading direction, propeller rates, thrust force and rudder angle.

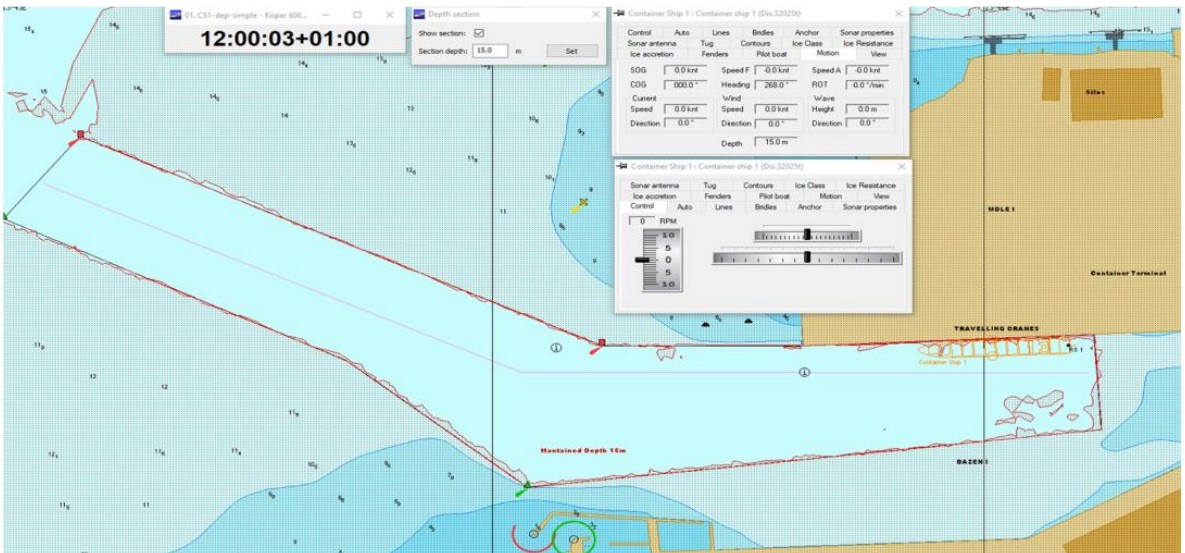

**Figure 3.** The investigated area shows the position of the outgoing container ship with a bathymetric depth of 15 m.

The thrust force ($T_e$) of the ship is then calculated in the simulator according to the following equation [10]:

$$T_e = C_t \left( J, \frac{p}{D_P} \right) \rho_w n^2 D_p^4 (1 - t) \tag{1}$$

Parameters in the equation are: thrust coefficient ($C_t$), which is a function of speed of advance ($J$), pitch to propeller diameter ratio ($p/D_P$), water density ($\rho_w$) propeller rotation per second ($n$) and thrust deduction coefficient ($t$). Speed of advance is calculated with the equation:

$$J = \frac{V}{nD_p} \tag{2}$$

Parameter ($V$) is vessel speed. The thrust deduction coefficient takes into account the underwater shape of the vessel. For conventionally-shaped single screw propelled vessels, it can be obtained with the following expression [11]:

$$t = 0.001979 \frac{L_{wl}}{(B - BC_{P1})} + 1.0585C_{10} - 0.00524 - 0.1418 \frac{D_p^2}{(Bd)} + 0.0015C_{stern} \tag{3}$$

The parameters in the equation are: length on the waterline ($L_{wl}$); ship's breadth ($B$); draft ($d$); coefficient ($C_{10}$), which for most of the ship is equal to the ratio between the ship's beam and waterline length, while coefficient ($C_{stern}$) varies from $-25$ to $10$ based on the shape of the ships stern; and coefficient ($C_{P1}$), which is a function of the prismatic coefficient and longitudinal center of buoyancy. The thrust was compared with the outflow velocity of the propeller jet at a nonzero ship speed ($V_{0;v \neq 0}$). The propeller rates recorded by the simulator are further processed using semiempirical methods in order to obtain the applied power ($P_{app}$) and to determine the propeller jet exit velocity and the jet velocity distribution behind the propeller.

*2.2. Semi-Empirical Equations to Calculate Propeller Jet Efflux Velocity and Velocity Distribution behind the Ship Propeller*

The propeller revolutions per minute ($RPM_{app}$) are recorded during ship manoeuvres with a time step of 2 s. The power applied can be calculated using the following equation:

$$P_{app} = \left( \frac{RPM_{app}}{RPM_{max}} \right)^3 P_{max} \tag{4}$$

where ($RPM_{max}$) is the maximum propeller revolutions per minute and ($P_{max}$) is the maximum installed engine power. The propeller jet velocity is normally calculated using Equations (9) and (10), but in some cases the thrust coefficient is not known, in which case the efflux velocity ($V_0$) can be obtained using the following expression [12]:

$$V_0 = C_3 \left( \frac{P_{app.}}{\rho_w D_p^2} \right)^{\frac{1}{3}} \tag{5}$$

Coefficient ($C_3$) is equal to:

- 1.17 for ducted propellers (propellers with nozzle);
- 1.48 for free propellers.

The equation above is used for ship bollard pull conditions (delivered by a ship when pulling at zero speed). During an actual manoeuvre, a ship has a certain speed (inertia moment) that must be considered:

$$V_{0;v \neq 0} = V_0 \left( 1 - \frac{V}{D_p n} \right) \tag{6}$$

Parameter ($V$) is vessel speed in (m/s). Jet efflux velocity for a vessel with a speed of more than zero is an important parameter for further analyses of maximum propeller jet bottom velocities ($V_{b,max}$), which determine critical sediment resuspension velocities of specific sediment structure [13]:

$$V_{b,max=0} = E V_{0;v \neq 0} \left( \frac{D_p}{h_t} \right)^b \tag{7}$$

where $E = 0.71$, $b = 1.0$ for seagoing ships with rudder; $E = 0.42$, $b = 0.275$ for seagoing ships without a rudder; $E = 0.52$, $b = 0.275$ for seagoing ships with a twin propeller and double rudder.

The distance between the propeller axis and the bottom of the sea is represented by the ($h_t$) value.

$$h_t = C + \frac{D_p}{2} \tag{8}$$

The clearance ($C$) is the distance between the propeller tip and the seabed. The equation gives the maximum velocity of the propeller jet at the seafloor and provides information about the manoeuvrability of the vessel in terms of a lower sediment resuspension rate. For a more accurate assessment of how much sediment is resuspended during the ship's manoeuvre, another method is needed that provides a distribution of jet velocity in the vertical and horizontal directions with respect to the propeller axis. There are several authors who have addressed this problem, such as [5,14]. Models are used for the same purposes.

A Schematic view separates the propeller jet velocity into three zones (Figure 4):

1. Efflux zone ($0 < X/D_p < 0.35$);
2. Zone of flow establishment ($0.35 < X/D_p < 3.25$), determined by [15];
3. Zone of established flow ($3.25 < X/D_p < 50$).

where parameter ($X$) presents horizontal distance behind the propeller.

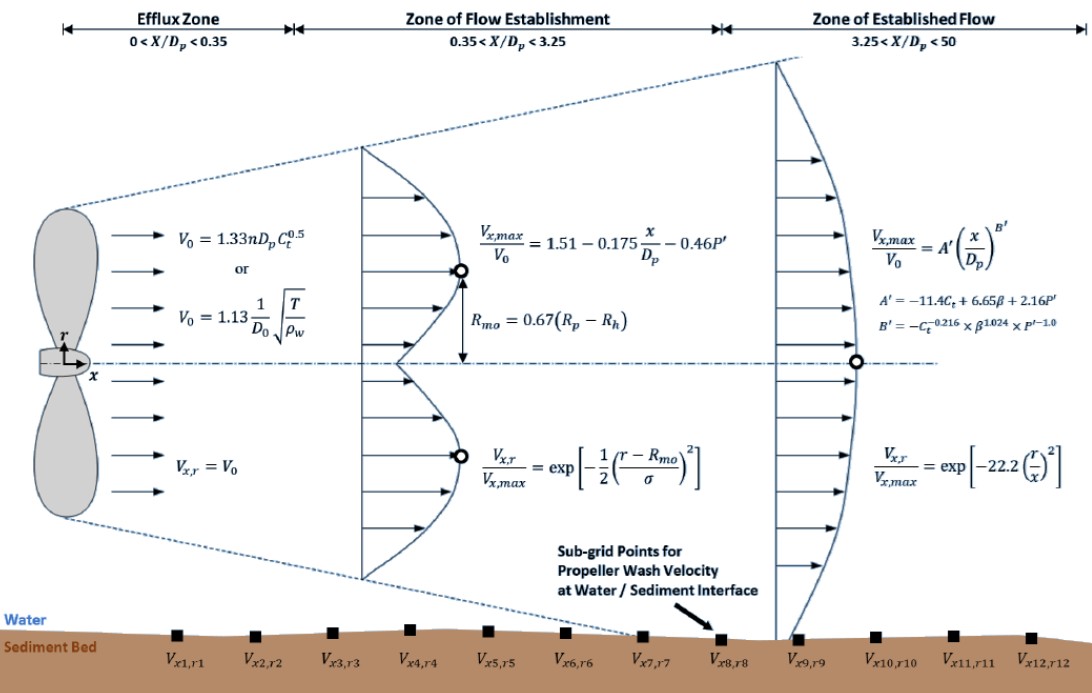

**Figure 4.** Propeller jet velocity distribution [16].

Propeller jet efflux velocity necessitates two equations. The first is based on axial momentum theory [17], taking into account thrust coefficient:

$$V_0 = 1.33 n D_p \sqrt{C_T} \tag{9}$$

The second equation [18] takes into account propeller thrust ($T$), water density ($\rho_w$) and propeller jet contraction ($D_0$). The equation is particularly useful when the FMBS simulation tool is coupled with MIKE 3 FM, which records dynamic ship data, among which is thrust force. It is used to determine the propeller outflow velocity, which is used for further jet velocity distribution.

$$V_0 = 1.13 \frac{1}{D_0} \sqrt{\frac{T}{\rho_w}} \tag{10}$$

The jet velocity field in Zone 1 is determined to be the same ($V_0 = V_{x,r}$) in longitudinal ($X$) and vertical ($r$) directions, up to the axial distance ($0 < X/Dp < 0.35$) from the propeller.

The maximal propeller jet efflux velocity decay in Zone 2 is determined using the equation proposed by [4]:

$$\frac{V_{x,max}}{V_0} = 1.51 - 0.175 \left( \frac{X}{D_p} \right) - 0.46 \frac{p}{D_P} \tag{11}$$

The propeller jet velocity field ($V_{x,r\_2}$) in Zone 2 was first determined by the Gaussian normal probability function [19]; later [20], this equation was improved and presented the jet velocity field in Zone 2. The equation is based on pitch to diameter ratio ($p/D_P$).

$$\frac{V_{x,r_2}}{V_{max}} = EXP \left( -\frac{1}{2} \frac{(r - R_{m0})^2}{\sigma^2} \right) \tag{12}$$

Parameter ($r$) determines the radial distance from the propeller axis. The author [20] measured standard deviation ($\sigma$) as a constant equal to 0.5 $R_{m0}$ up to the downstream distance of $X/D_p = 0.5$.

$$\sigma = \frac{1}{2} R_{m0} \; for \; \frac{X}{D_p} < 0.5 \tag{13}$$

Beyond $X/D_p = 0.5$ to the end of the zone of flow establishment, the standard deviation was defined as:

$$\sigma = \frac{1}{2} R_{m0} + 0.075\left(X - \frac{D_p}{2}\right) \; for \; \frac{X}{D_p} > 0.5 \tag{14}$$

The parameter ($R_{m0}$) is the radial distance of maximum velocity from the propeller shaft axis [21]. It is a function of the propeller radius ($R_p$) and propeller hub radius ($R_h$):

$$R_{mo} = 0.67\left(R_p - R_h\right) \tag{15}$$

The maximum propeller jet velocity in Zone 3 was determined by [20], who performed experimental measurements based on the propeller geometry characteristics, and proposed the following equation:

$$\frac{V_{max}}{V_0} = A'\left(\frac{X}{D_p}\right)^{B'} \tag{16}$$

Parameters ($A'$) and ($B'$) are determined using the following equations:

$$A\prime = -11.4 C_t + 6.65\beta + 2.16 P' \tag{17}$$

$$B\prime = -C_t^{0.216}\beta^{1.024}P'^{-1.87} \tag{18}$$

where the propeller blade area ratio is described with $\beta$.

The propeller jet velocity field ($V_{x,r\_3}$) in Zone 3 was determined according to [17], with the next equation being

$$\frac{V_{x,r\_3}}{V_0} = EXP\left[-22.2\left(\frac{r}{X}\right)^2\right] \tag{19}$$

The equations presented above are used to determine the velocity field of the propeller jet in the longitudinal ($X$) and vertical ($r$) directions corresponding to the propeller axis. The disadvantage of the presented semiempirical equations is that they do not take into account the distance between the propeller axis and the seabed, the bathymetry of the seabed, the deflection of the propeller jet by pier walls, the different depth zones in the ports and the shape of the harbour basins. A possible solution to these drawbacks is to recreate a ship's manoeuvre with the FMBS and find a suitable method to calculate the propeller jet efflux velocity. There are several computer programs based on Computational Fluid Dynamics (CFD): Environmental Fluid Dynamics Code plus (EFDC+), which is an open source three-dimensional (3D) finite-difference surface water modelling program. The CFD is a branch of fluid mechanics that deals with data structures and numerical analysis to solve flow problems; such programs have already been used by authors to present fluid and sediment motion caused by ship propeller rotation [22–27]. Maynord's model [28] and the Finite Analytical Navier-Stokes Solver (FANS model) [29] were interlinked with Curvilinear Hydrodynamics in three dimensions plus Tableau Input Coupled Kinetic Equilibrium Transport (CH3D + TICKET) [30]. It is a simulation tool to present contamination fate transport and redeposition of the sediment particles from the ship propeller jet. The CFD method is relatively inexpensive compared to experimental investigations. In this work, the hydrodynamic model MIKE 3 FM is coupled with MIKE 3 MT to represent the velocity distribution of the propeller jet corresponding to the bathymetry of the Koper Basin 1 harbour and mud transport after the propeller jet impedes the port bottom. The MIKE 3 FM is presented in the next subsection.

### 2.3. The Hydrodynamic Model MIKE 3

The three-dimensional, baroclinic model MIKE 3 FM is a commercial nonhydrostatic numerical modelling system designed for a wide range of applications in areas such as oceans, coastal regions, estuaries and lakes. It simulates unsteady three-dimensional flows, taking into account density variations, bathymetry and external influences such as metocean information, tides, currents and other hydrographic conditions. Features of relevance to this study's hydrodynamic model include: bed resistance, density variations, turbulence modelling, isolated sources and sinks, connected source/sink pairs, pier resistance, particle tracking and discharge calculations [3].

The hydrodynamic model MIKE 3 FM in general is based on a continuity equation, using Cartesian co-ordinates.

$$\frac{\partial u}{\partial x} + \frac{\partial v}{\partial y} + \frac{\partial w}{\partial z} = S \tag{20}$$

Parameters ($u$, $v$, $w$) present flow velocity components in the spatial domain ($x$, $y$, $z$).

Simulation MIKE 3 FM uses time integration of the shallow water and the transport equation of fluid/particles is performed using semi-implicit scheme. Using an explicit scheme, the time step interval must be selected, determining the Courant–Friedrich–Levy (CFL) number, which must be less than 1. The free surface is presented with the sigma-co-ordinate transformation approach. The parameters temperature of fluid and salinity are based on a general transport–diffusion equation [31].

Ship propeller jet velocity distribution is based on integral jet model equations, presenting steady state solutions of jet by solving a conservation equation for flux and momentum, temperature and salinity [32].

MIKE 3 Input Data

High-resolution bathymetry within the study area is established as a first step (Figure 5). The hydrodynamic model MIKE 3 FM includes subprogram MIKE Zero with additional features such as: Time Series and Mesh Generator, which are used to represent the sediment resuspension process in this work. The Mesh Generator is used to define the bathymetry; the data input was done using the cartesian co-ordinate system ($x$, $y$, $z$ file). The bathymetry was defined as follows: maximum element area 8 m$^2$, minimum allowed angle 26° and maximum number of nodes 55,000.

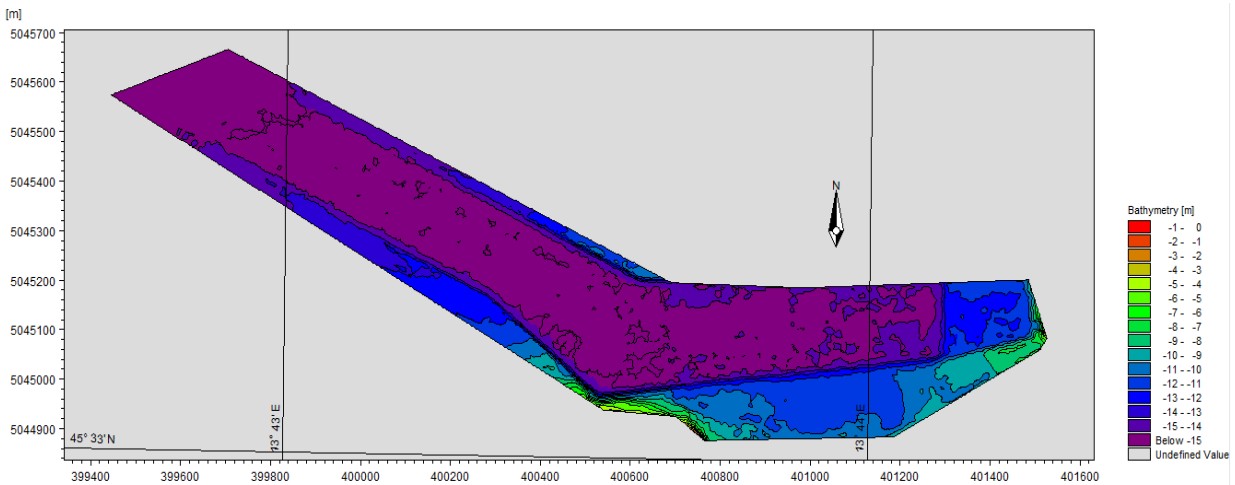

**Figure 5.** MIKE Zero Mesh Generator; Port of Koper, Basin I.

The second step is to specify bathymetry boundary conditions. Figure (Figure 6) shows the red zone which is specified as the fluid free outflow condition and green zone specifying land as containment zone (zero normal velocity).

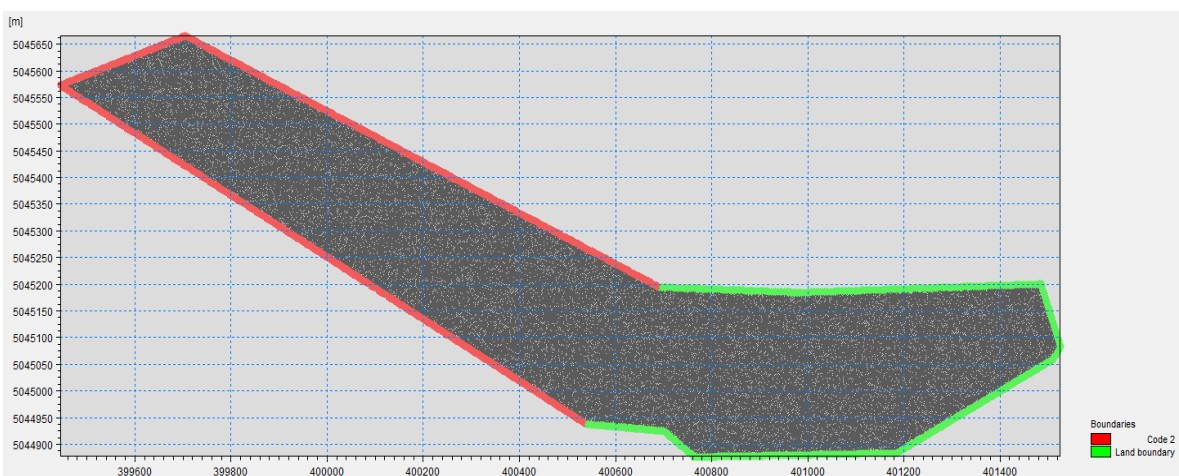

**Figure 6.** Bathymetry boundary conditions specification.

The departure manoeuvre on FMBS was performed; ship motion data were recorded. The vessel ($RPM_{app}$) is used in Equation (4) to obtain ($P_{app}$), which was needed to calculate (Equation (5)) the propeller jet efflux velocity for zero ship speed, and later used in Equation (6) to obtain the propeller jet efflux velocity for nonzero ship speed. MIKE Zero contains time series'; the input data are: Discharge (m$^3$/s), U-velocity component (m/s) and *V*-velocity component (m/s). The ship's propeller is determined as a nozzle; the discharge rate (depending on the propeller diameter and efflux velocity when the ship velocity is non-zero), the *U*-velocity, *V*-velocity depend on the ship's heading, the rudder angle and are calculated using the following equation.

$$Discharge = A_{prop}.V_{0;v\neq0} \tag{21}$$

$$U_{vel.} = sin\left(\frac{\alpha_{head;rudd}.\pi}{180}\right)V_{0;v\neq0} \tag{22}$$

$$V_{vel.} = cos\left(\frac{\alpha_{head;rudd}.\pi}{180}\right)V_{0;v\neq0} \tag{23}$$

Propeller surface ($A_{prop}$.) and angle ($\alpha_{head;rudd}$) from geographic north, taking into account the ship's heading and rudder angle ($\alpha_{head;rudd}$). The results are inserted in MIKE 3 FM as a standard source (varying in time). The ship's departure route was determined using 17 sources/locations, as shown in the following figure (Figure 7).

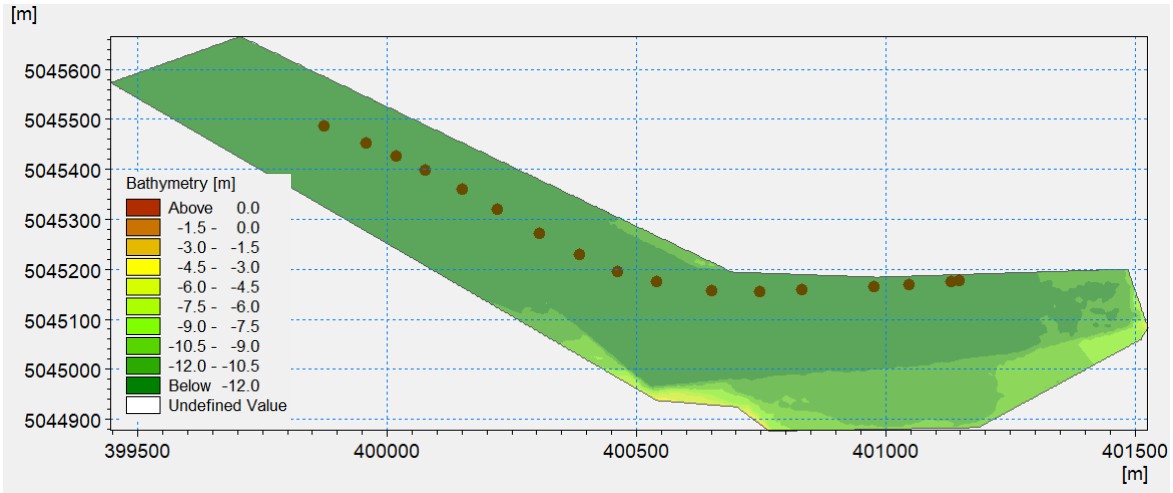

**Figure 7.** Propeller jet source determined by 17 locations.

Each source is determined by Equations (19)–(21) in the variant time domain. The final source is the position of the ship when the simulation is stopped on FMBS (8.5 min into the ship's manoeuvring).

The simulation took 30 min to find out how the velocity field of the propeller jet evolves after a ship manoeuvre due to the variable bathymetry and characteristics of the basin. The sediment structure in the Koper port was determined using the following parameters: density of sediment 2650 (kg/m$^3$), bottom roughness 0.1 m and critical shear stress 0.1 (N/m$^2$). The overall modelling methodology is described in figure below (Figure 8).

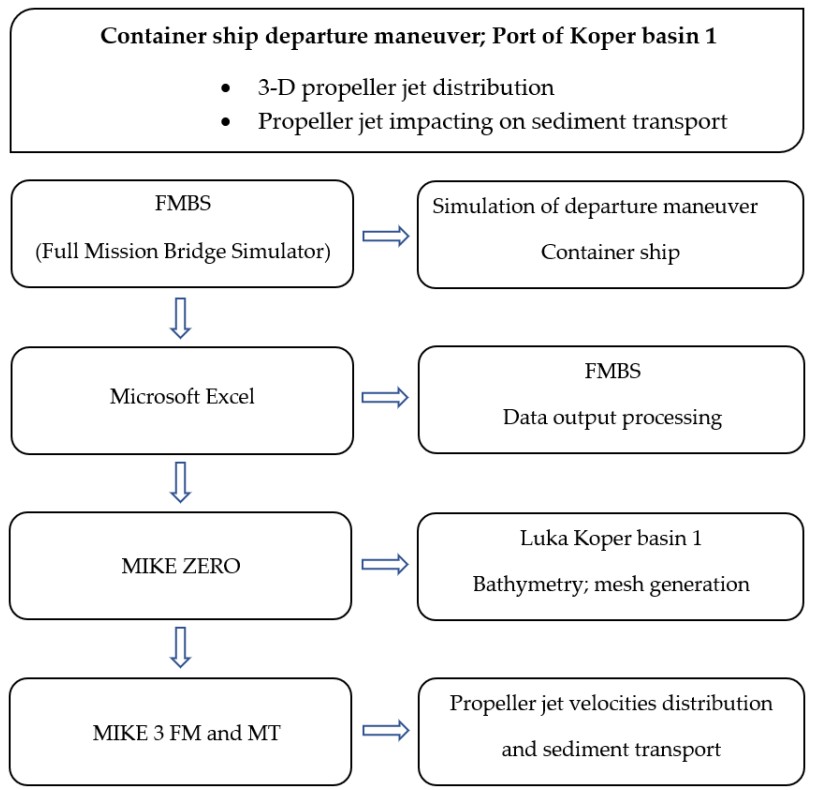

**Figure 8.** Research methodology review.

## 3. Results and Discussion

The results of the hydrodynamics and sediment transport modules are presented in this chapter. The numerical domain was vertically calculated at ten different depths. The kinematics of the vessel and the efflux velocities are illustrated in Figure 9. The most important parameters are propeller rates per minute ($RPM$), propeller efflux velocity ($V_0$), propeller efflux velocity for a nonzero ship speed ($V_{0;v \neq 0}$) and ship speed ($SOG$) in relation to time (8.5 min). The ship speed increases constantly from 0 to 6 m/s over 8.5 min. The departure manoeuvre starts with the command "Dead Slow Ahead", resulting in the increased propeller flux velocity($V_0$), which is extremely invasive as the vessel is stationary, and as the vessel begins to move the efflux velocity slowly declines calculated by the nonzero ship speed ($V_{0;v \neq 0}$). Initially, efflux velocity is 6 m/s with the discharge rate of 350 m$^3$/s. When the ship's speed is increasing, efflux velocity is decreasing (black line).

After about 5 min of manoeuvring, the command "slow ahead" is given; a significant increase of ($V_0$) 8.5 m/s and a decrease of ($V_{0;v \neq 0}$) 4.7 m/s are observed. The latter is the main parameter influencing mud steering-up process. At the beginning of the manoeuvre, the propeller slip is considerable; resulting in a high ($V_{0;v \neq 0}$), with increasing (SOG) and steady (RPM). The conclusion is that the decrease in ($V_{0;v \neq 0}$) depends mainly on the relationship between ($SOG$) and ($RPM$).

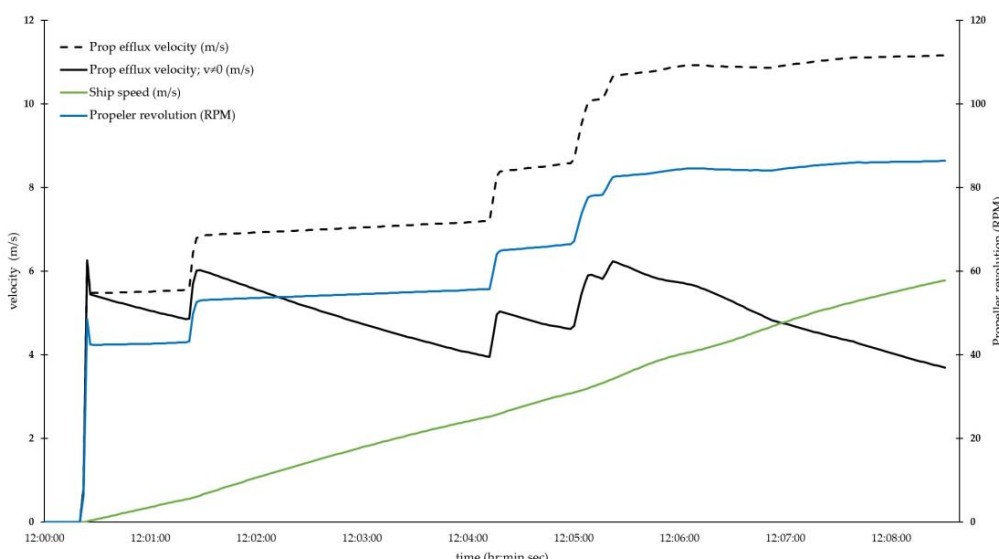

**Figure 9.** Ship propulsion jet velocity characteristics.

Previous research regarding bottom wash assessment in the Port of Koper has been presented by [33]. In this study, the authors determined the critical propeller jet bottom velocity of 0.35 m/s, at which the jet velocity overcomes the friction of the mud bottom and causes the detachment of sediment particles and their circulation from the seafloor. Figure 10 shows the jet velocity (m/s) generated by the ship's propeller in bottom layer 1. Figure 11 shows the resulting suspended sediment concentration (kg/m$^3$) in the same layer. The caption is from ship positions 1 and 2, respectively (Figure 7).

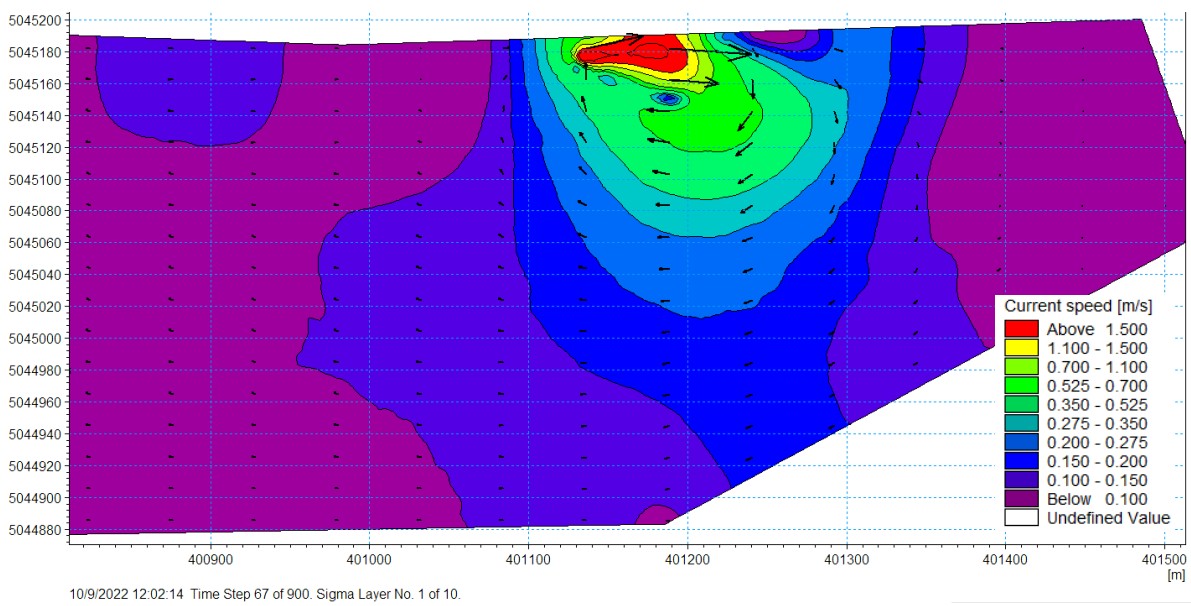

10/9/2022 12:02:14 Time Step 67 of 900. Sigma Layer No. 1 of 10.

**Figure 10.** Ship propeller generated jet in bottom layer (m/s).

Ship propeller generated jet in bottom layer (m/s) Figure 10 shows the corresponding velocity fields of the propeller jet at 134 s when the sediment particles coloured green, yellow and red detach from the bottom. The vectors show the direction and velocity of the propeller flow. The flow is deflected to the right about 100 m ahead of the propeller front due to the starboard rudder angle (STBD) of 11 degrees, the proximity of the pier wall, the horseshoe-shaped port depression and the decrease in port depth from 15 m to 13 m at position long. 401,300. A nearly negligible contribution to the current deflection is

due to the Coriolis force, which forces the fluids in the northern hemisphere to turn in the clockwise direction.

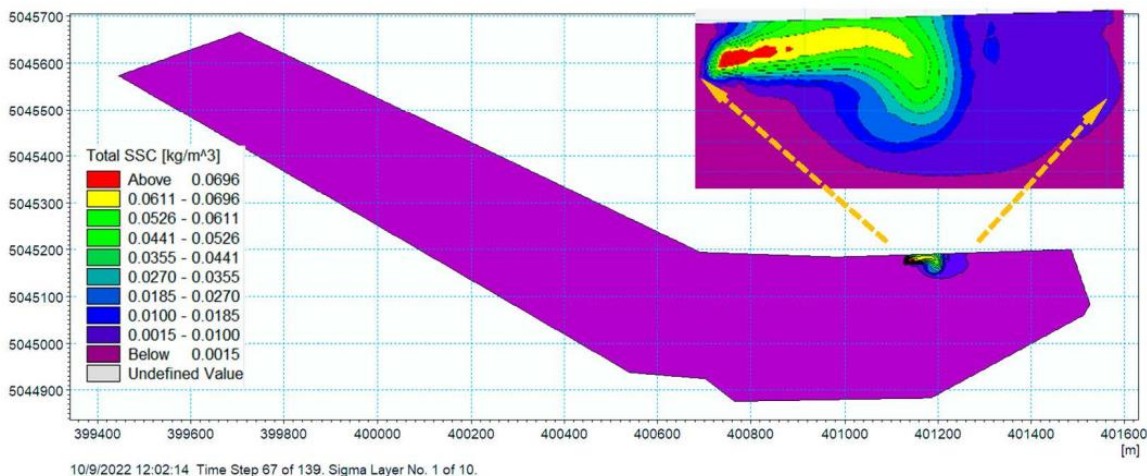

**Figure 11.** Resulting suspended sediment concentration (kg/m$^3$).

Figure 11 shows the total specific sediment concentration (SSC) in (kg/m$^3$) in bottom layer 1; during investigations, it can reach a maximum value of 0.07 (kg/m$^3$). The resuspended sediment particles are further advected by the flow field induced by the propeller jet (Figure 10).

The propeller jet-induced flow and total (SSC) are shown in Figure 12. The duration of the manoeuvre is 510 s. The figures were taken after 1000 s; they show the flow velocity/path and the total (SSC) in bottom layer 1.

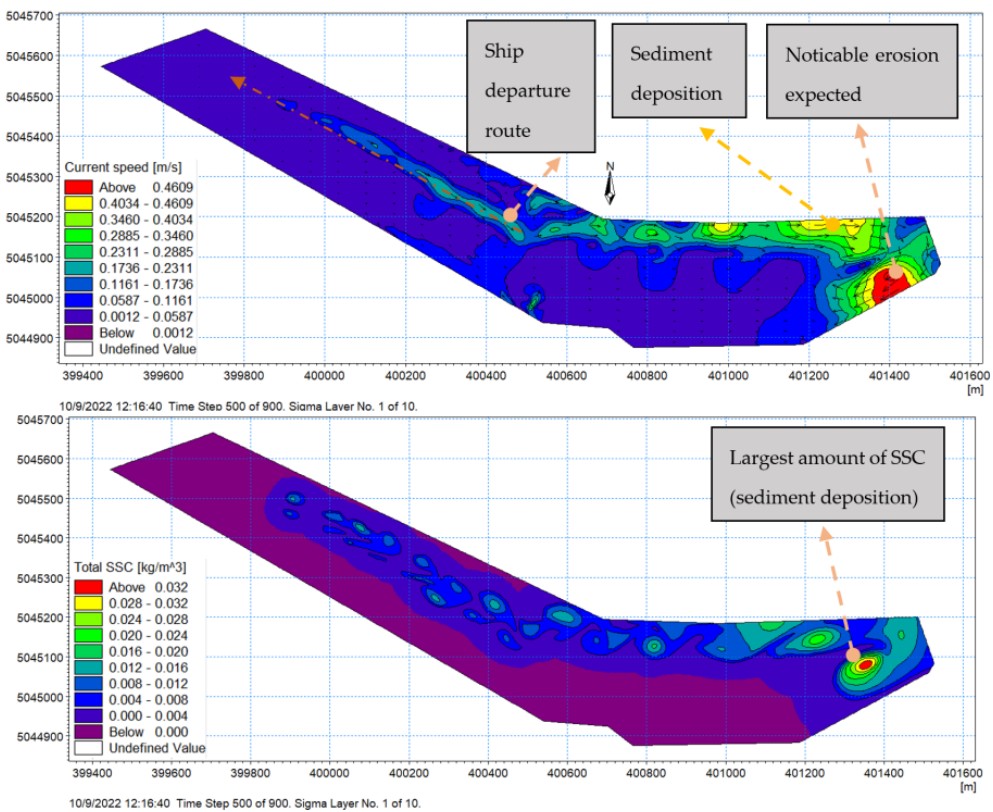

**Figure 12.** Current velocity/path and total SSC in bottom layer 1.



As expected, there is a significant decrease in flow velocity (by a factor of three: see Figure 12). From the colour palette, the ship's departure route can be clearly seen. The flow is deflected clockwise due to the horseshoe-shaped basin. The south-eastern part of the basin is subject to significant erosion of the sediment material, which is influenced by the high velocity flow field.

The low velocity flow eddy surrounded by high velocity fields causes sediment material deposition, marked on Figure 12. The second part of Figure 12 shows the SSC in the same bottom layer and the time step. The colour palette shows the content of resuspended material. The largest amount of SSC is expected at the position shown in the first part of the figure (sediment deposition). Sediment accumulation is expected at the same position.

Validation of the MIKE 3 MT simulation program was performed using a satellite image of Basin 1 from a similar vessel. Figure 13 shows the SSC at the surface (layer 10). It can be seen that the largest sediment concentration is formed in a vortex position at the same location as in the real ship scenario (Google Earth image).

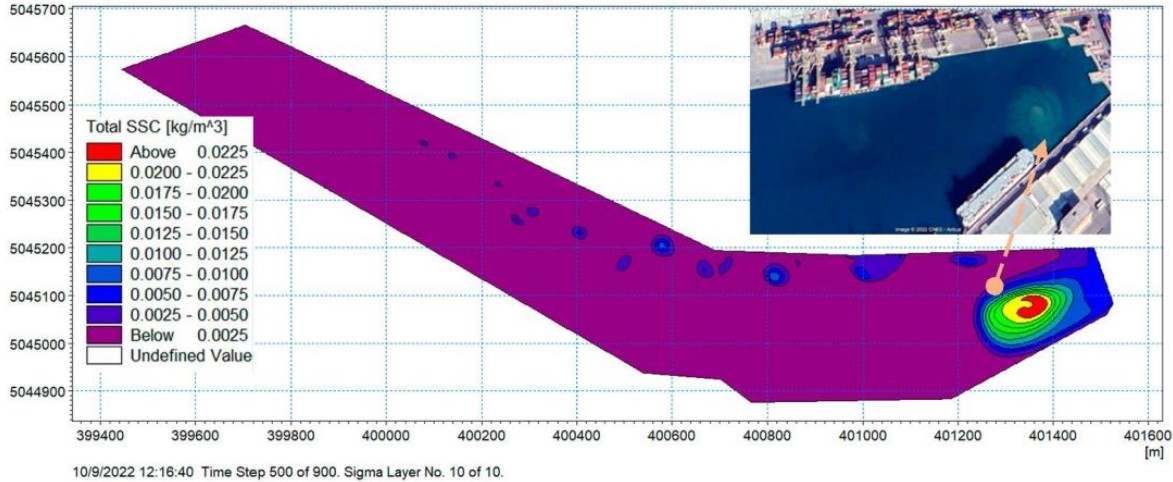

**Figure 13.** Sediment concentration (SSC) in surface layer and validation with satellite picture.

Validation of results is also based on measurements performed 25 July 2013 in the bay of Koper at 38 measurement locations after the departure manoeuvre of the container ship Ital Universo (IMO no.: 9196993). The determination tool of total resuspended particles (SSC) has been water samples, taken at three depths (0.5, 6.0 and 12.0 m). Measurements have been performed after 3 min., when the tugs' lines have been released. The paper [34] considers measurements from the manoeuvre zone (Figure 14), where it was displayed as a maximum value of (TSS~0.139 $kg/m^3$) and sampled a concentration of (SSC~0.037 $kg/m^3$) in the bottom layer.

Comparing the numerical modelling with the experimental investigations above, there was clearly impressive similarity in the results. Both modelling and sampling were performed in the bottom layer (hydrodynamic model at 1.0 m; and in situ sampling at 0.5 m). The hydrodynamic model showed (Figure 11) noticeable (SSC~0.07 $kg/m^3$) after 2 min 14 s from the start of the manoeuvre. The container ship was first towed astern from berth and after being turned, the vessel accelerated with power similar to that in our case. Their measurements of total resuspended particles were similar (SSC~0.037 $kg/m^3$).

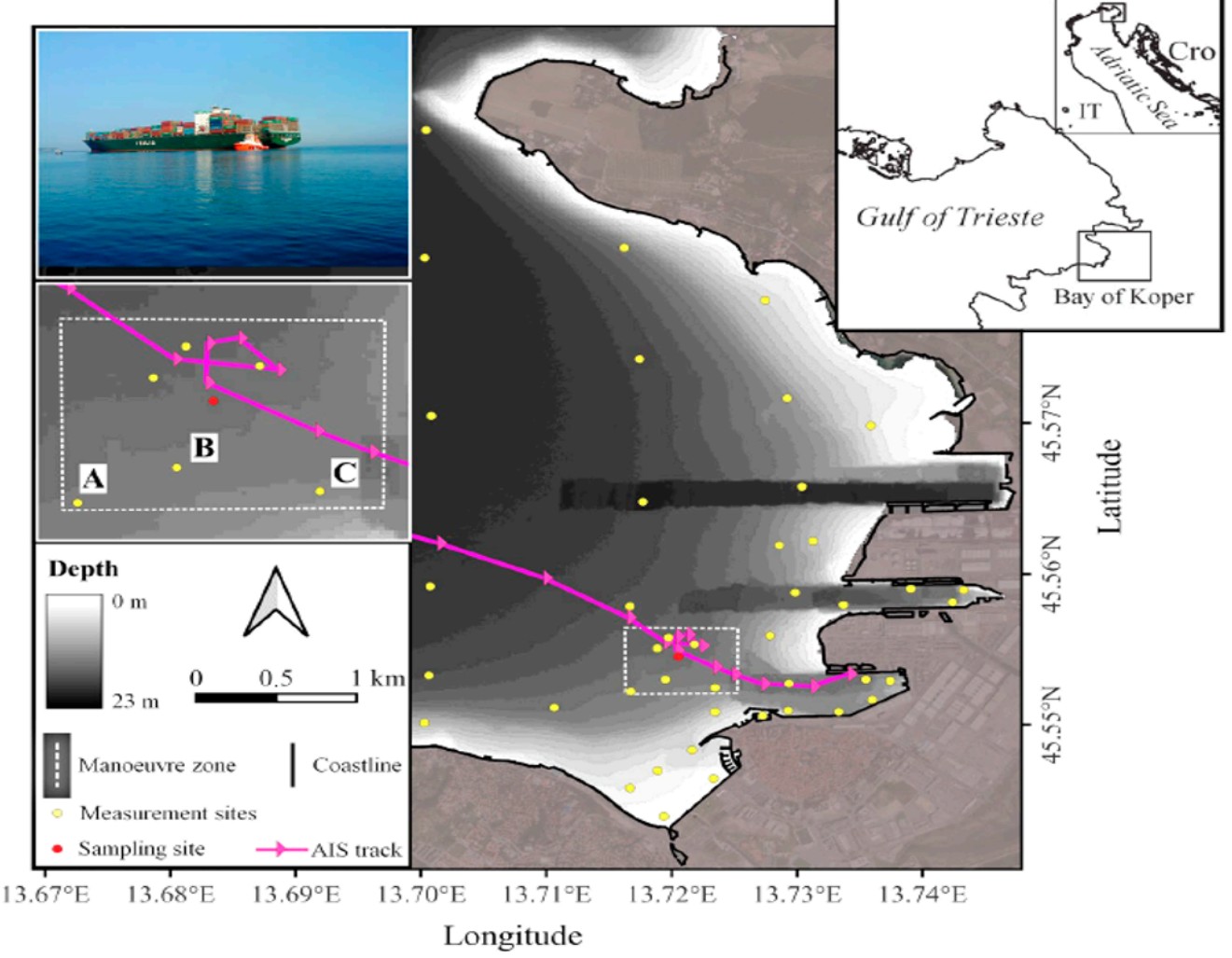

**Figure 14.** Bay of Koper. The yellow marks presenting 38 measurements locations and purple line presents departure route from container terminal Basin 1, letters A, B, C presenting prior measurement to verify content of resuspended particles prior to the vessel manoeuvre [34].

The difference in (SSC) between the real case and the modelled case is mainly due to the greater depth of the former and the presence of two tugs.

The significance here is that now that we are reassured that our model successfully mimics actual circumstances, we can proceed to examine any circumstances, any ship size, UKC, propeller type, and so on, and determine the optimal manoeuvring approach. We are confident as well that adding tugs and thrusters will not have the slightest deleterious effect on the modelling. Perhaps it should also be mentioned that as was implied early on, the effect of an abstract (having nothing to do with nature) decision-making motive and mindset is liable to continue to have a degrading effect on port areas and beyond, and we will have a relatively easy means of displaying these effects. Of course, some protective measures have been introduced, such as the rip-rap technique, and we will be able to monitor their effectiveness [35,36]. Based on our calculation, we can recommend a rip-rap protection area in the northern part of Basin 1 along the entire container terminal and 40 m in lateral distance (Figure 15).

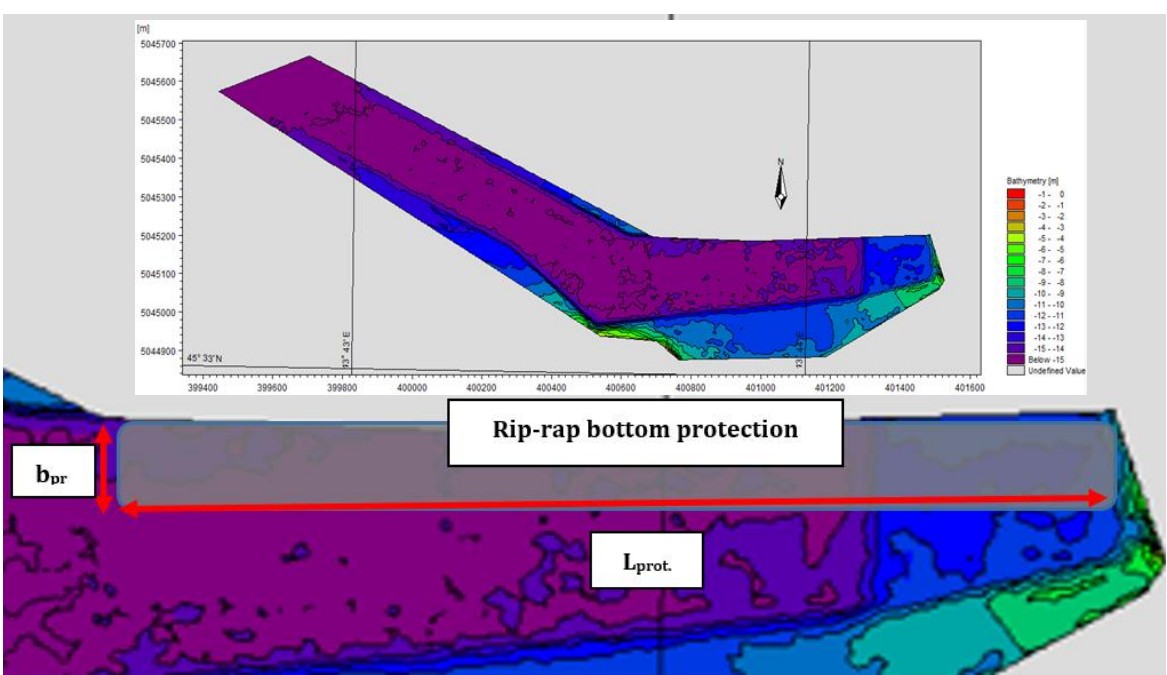

**Figure 15.** Port of Koper Basin 1; recommended bottom protection area [37].

## 4. Conclusions

The impact of ship traffic on the seabed of Koper Harbour Basin I was analysed using hydrodynamic numerical models based on a high-resolution non-hydrostatic circulation model (MIKE 3 HD FM) with a sediment transport model (MIKE 3 MT FM). The simulation results show that the ship propeller jet can reach velocities of more than 1.5 m/s. The intensity of the ship propeller jet depends on the following factors: propeller dimensions and RPM, distance between the propeller and bottom and whether the manoeuvre is a departure or an arrival. The critical jet bottom velocity at which sediment particles detach from the seafloor is reported to be 0.35 m/s. The jet zone of the propeller can reach distances up to 30–40 times the propeller diameter in the horizontal direction. The simulation allowed validation of the sediment dynamics in the harbour using satellite images of the sea surface showing the generated sediment eddy (Figure 13). Further simulation analysis shows why certain port areas are subject to sediment erosion and others to deposition. A nonhydrostatic circulation model (MIKE 3 HD FM and MT) was tested to determine its ability to represent ship-induced propeller-generated jets. It was determined that the tested programme is suitable for further research, replicating alternative ship manoeuvres using the Full Mission Bridge Simulator and later hydrostatic ship motion data implemented in the MIKE 3 HD FM fluid motion program.

The limitation of this paper aligns well with our intent: now that our model is utile, we intend to use this model to conduct a number of applications and find the optimal means of manoeuvring in the port under all variety of circumstances.

**Author Contributions:** Conceptualization, Principles and methods, Conducting surveys, Data analysis, Software and Conclusions done by J.S., Principles and methods and Conducting surveys, Validation done additionally by M.P. and A.G. All authors have read and agreed to the published version of the manuscript.

**Funding:** The publication of the paper is partially financed by the research group at the Faculty of Maritime Studies and Transport, financed by the Slovenian National Research Agency.

**Institutional Review Board Statement:** Not applicable.

**Informed Consent Statement:** Not applicable.

**Data Availability Statement:** The data sets used and/or analyzed in this study are available from the corresponding author upon reasonable request.

**Acknowledgments:** The authors acknowledge the Port of Koper, the Slovenian Maritime Administration, marine pilot Matjaž Felicjan, and Antonio Guarnieri.

**Conflicts of Interest:** The authors declare no conflict of interest.

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
