# Peer review of "Sediment Resuspension Distribution Modelling Using a Ship Handling Simulation along with the MIKE 3 Application"

_jmse, doi:10.3390/jmse11081619_

Round 1

Reviewer 1 Report

The manuscript is about the effects of ship propeller jet on sediment resuspension and transport in ports, and the authors use a combination of ship handling simulator, semi-empirical equations, and numerical models to study this problem. On the whole, the manuscript is well written and in line with the research field of Journal of Marine Science and Engineering, but it could be improved in some aspects. Here are some suggestions for further evaluation:

1.       In the abstract, the authors should provide a brief summary of the main findings and contributions of your work, not just the methods and tools the you used. For example, how the model-based methodology can help optimize ship manoeuvres or port design to minimize these effects.

2.       The “introduction section” is too brief, and the reader will lose a lot of important information. The authors should provide more background information and context for your research problem, such as why sediment resuspension and transport in ports is an important issue, what are the existing challenges and gaps in the literature, and what are the objectives and scope of your work. And more recent and relevant references to support your claims and show the novelty and contribution of your work are needful.

3.       In section 2.1, Please explain why you chose the port of Koper, Basin I, berth 7 a as the authors case study, and what are the characteristics and conditions of this port that make it suitable for your analysis.

4.       The authors should also explain how the authors coupled the FMBS simulation data with the MIKE 3 FM model to represent the propeller jet source as a standard source varying in time.

5.       More details about the validation of their numerical models are missing, such as the error analysis, and the comparison with other existing models or experimental data.

6.       Please highlight how your findings can be used to optimize ship manoeuvres or port design to minimize sediment resuspension and transport, and what are the implications for port management, environmental protection, and navigation safety.

7.       In section 4, the authors should provide a concise summary of your main findings and contributions, as well as some suggestions for future work or implications for practice. Also please acknowledge any limitations or uncertainties of this approach and discuss how they could be addressed or mitigated.

Author Response

Reviewer 1

Dear Reviewer,

We would like to thank you for reviewing our manuscript. We greatly appreciate your efforts and your constructive and helpful feedback. In our recent version, we made a considerable effort to include this feedback adequately. All points raised by the reviewers were discussed amongst the authors and considered for revision. Based on your suggestion and the collective recommendations of the other reviewers, the following major issues were addressed:

The title was changed, we think for the better, which another reviewer suggested.

The abstract has been expanded, rewritten parts of the paper, generally with a view toward clarity but also with the attempt to add context to the significance of using this environmentally oriented research. Two figures were introduced to clarify the modelling approach.

The text was substantially reviewed by a native English speaker and a great many changes were necessary, so we thank you for that as well, and would like to applaud your patience.

Suggested references are taken into account.

In regard to your detailed review, we are sending the paper with tracked changes, and you will see that we took into consideration all of your suggestions and remarks.

Reviewer 2 Report

The impact of ship wake flow on the seabed of Koper Harbour Basin is analyzed using hydrodynamic numerical models. The propeller velocity data are obtained basing on empirical formulas, and the sediment transport behaviors are analyzed under the departure manoeuver. The method is appropriate with high efficiency, and could extends to other scenarios. The following issues should be considered before it is accepted:

1.      The title needs to be revised to reflect on the major scenario (departure from the harbour) in current study.

2.      Section 2.2, the clearance C between the propeller tip and the seabed is a determined factor for the erosion and sediment. Whether the author considers exploring the law of seabed behavior under different clearance ? In addition, above changes can also be directly related to the load and draft of the ship.

3.      Line 260: It is difficult to read Graph 1. Different Y ordinates for every plot should be added at least.

4.      Satellite maps should be cited in Figure 4 and Figure 10.

5.      The format of references should be consistent according to the journal requirements.

Author Response

Reviewer 2

Dear Reviewer,

We would like to thank you for reviewing our manuscript. We greatly appreciate your efforts and your constructive and helpful feedback. In our recent version, we made a considerable effort to include this feedback adequately. All points raised by the reviewers were discussed amongst the authors and considered for revision. Based on your suggestion and the collective recommendations of the other reviewers, the following major issues were addressed:

The title was changed, we think for the better.

The abstract has been expanded, rewritten parts of the paper, generally with a view toward clarity but also with the attempt to add context to the significance of using this environmentally oriented research. Two figures were introduced to clarify the modelling approach.

The text was substantially reviewed by a native English speaker and a great many changes were necessary, so we thank you for that as well, and would like to applaud your patience.

Format of the used references are considered.

Presentation of the Graph 1 is improved.

In regard to your detailed review, we are sending the paper with tracked changes, and you will see that we took into consideration all of your suggestions and remarks.

Reviewer 3 Report

In this manuscript, a numerical study was aimed at exploring the effects of ship propeller jet velocity field distribution on the seabed of the Koper Harbour basin. 

The paper is well-organized and concise. The specific needs of each stage of development are identified and well outlined. Therefore, I believe that this manuscript may be of interest to JMSE readers and I would recommend that the paper be accepted after addressing some minor comments described below.

* I strongly suggest the authors revise the manuscript to meet the journal structure standard, edit the English language, and most of all rework the accuracy of symbols in the equations and text. 

Major issues are highlighted hereafter. 

- Line 58: Table 1. Please replace "ship type" with "ship characteristics".

- Line 62: The acronym FMBS (Full Mission Bridge Simulator) has to be defined in the text. 

- Line 69: Please replace "thrust coefficient (K)" with "thrust coefficient (Kt)". 

- Line 78: The symbol "T" was used indifferently for (a) draft, (b) propeller thrust (line 133), and (c) thrust force (line 135). Please clarify. 

- Line 80: I suggest not mentioning the coefficient "Cp" as it is never used in the equations and text. 

- Line 81: Longitudinal center of buoyancy (LCB). See the comment above.  

- Line 88: The parameter "Propeller rates per minute" (RPMapp) has been already defined (see line 83).

- Line 93: The authors mention: "In some cases thrust coefficient (Ct) is not known". The same coefficient has been defined also in lines 131 and 164. Please clarify. 

- Line 94: "that’s way [6] presented..." OR "that’s why [6] presented...". Please clarify. 

- Eq. (16), line 160. The exponent "B" has been introduced before as "ship's breadth (B)" (see line 78). 

- Eq. (18), line 167. This must be changed in Eq. (19). Please renumber. 

- Line 184: Modify "he three-dimensional" with "The three-dimensional". 

- Line 286: I believe you are referring to "Coriolis force". Please modify. 

- Line 347: "MIKE 3 HD FM soft-wear". I believe you are referring to "MIKE 3 HD FM software". Please correct. 

* Although the references provided are appropriate, in this field, there are many recent works worth mentioning. Some suggested examples are: 

- Penna, N., D’Alessandro, F., Gaudio, R., Tomasicchio, G.R. (2019). Three-dimensional analysis of local scouring induced by a rotating ship propeller. Ocean Engineering, Elsevier, 188, 106294.  

- Wei, M., Chiew, Y.M. (2018). Characteristics of propeller jet flow within developing scour holes around an open quay. J. Hydraul. Eng. 144 (7), 04018040.

* In general, the author often uses non-standard English that should be revised by an English native speaker.

* In general, the author often uses non-standard English that should be revised by an English native speaker.

Author Response

Reviewer 3

Dear Reviewer,

We would like to thank you for reviewing our manuscript and we greatly appreciate your efforts and your constructive and helpful feedback. An extensive revision was conducted on the original submission. Based on your suggestion and the collective recommendations of the other reviewers, the following major issues were addressed:

The title was changed, we think for the better, which another reviewer suggested.

The abstract has been expanded, rewritten parts of the paper, generally with a view toward clarity but also with the attempt to add context to the significance of using this environmentally oriented research. Two figures were introduced to clarify the modelling approach.

The text was substantially reviewed by a native English speaker and a great many changes were necessary, so we thank you for that as well, and would like to applaud your patience.

Suggested references are taken int account.

In regard to your detailed review, we are sending the paper with tracked changes, and you will see that we took into consideration all of your suggestions and remarks.

Reviewer 4 Report

The Authors present the results of simulations of sediment resuspension induced by a passing ship.

In my opinion, the manuscript appears as a technical report of the application of the software rather than a scientific article. Unfortunately there is not enough scientific content and novelty for the submission to be considered suitable for publication. The presentation of results should be much improved and many aspects should be clarified. I suggest that it is clearly stated what content is new and possibly compare the results obtained by the software with CFD results or experiments.

Detailled review

lines 47-49: More information and possibly references should be added regarding the state-of-the-art in the field of the evaluation of the propeller jet thrust

line 52: what is Automatic Identification System? Non experts in the field should be able to understand. Please either explain or introduce a reference

line 53: please clearly state what parameters were measured

table 1: columns in the table are arranged incorrectly

line 62: No details of this software are provided. Is it commercial or In-house developed? Please elaborate

line 83: The fact that a specific software was used does not appear to be relevant to the reader

 line 175: this descritpion is not complete

lines 179-181: I suggest to reduce the section on empirical methods from literature. This is confusing to the reader and a clear statement about what is actually new and what is state-of-the-art should be inserted. 

line 184: much more information about MIKE software should be provided. Is this open-source or commercial? 

line 188: Features of relavant Paper's hydrody-dynamic model include: this sentence is not clear

line 195: please explain what is MIKE Zero

No comparison is provided to either experimental results or simulations, except for the qualitative comparison provided in figure 10. For this reason it is difficult to assess the validity of the approach presented.

There are many typos and terms which should be corrected.

51: for the determination

line 68: D is without suffix

line 69: t suffix missing, parenthesis misplaced

line 94: that’s way [6] presented propeller

line 99: Equation valids for ship bollard pull condition.

line 184: he

line 227: Propeller orifface surface

line 329

line 347: soft-wear

Author Response

Reviewer 4

Dear Reviewer,

Naturally we thank you for your attention to detail and your suggestions. Most difficult was the fact that in part the paper IS a technical report on the use of the software. To our minds, however, we have advanced the use of not only the ship handling simulation, but introduced a method combining this with another tool.

The abstract has been expanded, rewritten parts of the paper, generally with a view toward clarity but also with the attempt to add context to the significance of using this environmentally oriented research. Two figures were introduced to clarify the modelling approach.

Of more value than anything new, to our minds, is the notion of comparing results with those of a study conducted by a nearby marine biology station. We were lucky to have at our disposal a study of the same problem in the very same space, basin 1 of the port in Koper.

The title was changed, we think for the better, which another reviewer suggested.

The text was substantially reviewed by a native English speaker and a great many changes were necessary, so we thank you for that as well, and would like to applaud your patience.

In regard to your detailed review, we are sending the paper with tracked changes, and you will see that we took into consideration all of your suggestions and remarks.

Round 2

Reviewer 4 Report

The Authors have addressed most of the concerns raised in the first version.

I would suggest that the Authors also include a point-by-point reply/rebuttal to the issues which were raised in the first version. This would make the review process faster, especially since the changes to the first version of the manuscript are extensive. 

Author Response

Dear reviewer, attached you can find requested point by point response.

Thank you.
